# Online Relational Inference for Evolving Multi-agent Interacting Systems

**Beomseok Kang, Priyabrata Saha, Sudarshan Sharma, Biswadeep Chakraborty, Saibal Mukhopadhyay**
Georgia Institute of Technology
`{beomseok,smukhopadhyay6}@gatech.edu`

## Abstract

We introduce a novel framework, Online Relational Inference (ORI), designed to efficiently identify hidden interaction graphs in evolving multi-agent interacting systems using streaming data. Unlike traditional offline methods that rely on a fixed training set, ORI employs online backpropagation, updating the model with each new data point, thereby allowing it to adapt to changing environments in real-time. A key innovation is the use of an adjacency matrix as a trainable parameter, optimized through a new adaptive learning rate technique called AdaRelation, which adjusts based on the historical sensitivity of the decoder to changes in the interaction graph. Additionally, a data augmentation method named Trajectory Mirror (TM) is introduced to improve generalization by exposing the model to varied trajectory patterns. Experimental results on both synthetic datasets and real-world data (CMU MoCap for human motion) demonstrate that ORI significantly improves the accuracy and adaptability of relational inference in dynamic settings compared to existing methods. This approach is model-agnostic, enabling seamless integration with various neural relational inference (NRI) architectures, and offers a robust solution for real-time applications in complex, evolving systems. Code is available at `https://github.com/beomseokg/ORI`.

## 1 Introduction

Multi-agent interacting systems have been studied in various fields, including particle-based physical simulations [1–3], traffic systems [4, 5], and social networks [6–8]. Interaction among agents is crucial information to accurately model such systems and also provides interpretability in agent behaviors [9]. However, external observers can only access the trajectory of agents without knowing interaction graphs. Consequently, identifying unknown interaction graphs from observable trajectories of agents has been emerged as a specific problem referred to as relational inference [10].

In recent years, neural relational inference (NRI) and its variants have shown promising performance in synthetic and real-world environments [11–16]. Prior studies mostly aimed to present better network for NRI based on variational autoencoder (VAE) built with graph neural networks (GNN) [14, 15, 12, 17]. These methods involve an encoder to infer an interaction graph as an adjacency matrix from observed trajectories and a decoder to predict the future trajectories employing the inferred interaction graph. They generally perform training offline assuming the well-aligned distribution in training and testing data. Unfortunately, such assumption is frequently violated in practice due to the shifts in test condition, including sudden changes in the interaction, evolving system parameters and even dynamics itself. Building a relational inference model generalizable to all the different scenarios is challenging [18–20]. In this case, online learning is an attractive approach to continuously adapt the model to newly observed environments [5]. However, online learning for relational inference has been rarely explored.

38th Conference on Neural Information Processing Systems (NeurIPS 2024).

Table 1: Comparison of key features between prior works and this work.

| Method | Description | Model Agnostic | Consider Evolution in | | | Criteria | |
|---|---|---|---|---|---|---|---|
| | | | Interaction | Parameter | Dynamics | Acc. | MSE |
| Prior offline works [11, 7, 12, 23, 16] | · offline backpropagation · novel encoder and decoder | - | ✓ | × | × | ✓ | ✓ |
| Prior online work [5] | · online convex optimization · constant learning rate | × | ✓ | × | × | × | ✓ |
| This work | · online backpropagation · AdaRelation; Traj. Mirror | ✓ | ✓ | ✓ | ✓ | ✓ | ✓ |

✓ indicates the presence of a feature, and × indicates the absence of a feature.

Online backpropagation using gradient descent is a widely used online learning method as it is compatible with various neural network designs [21]. However, online backpropagation on existing NRI models significantly degrades the accuracy on relational inference, since their decoder quickly learns the trajectory prediction even before the encoder generates reasonable interaction graphs. It is important to note that while the models are trained in self-supervised manner (*i.e.*, trajectory prediction), true labels of interaction graphs are never provided to the models, indicating that identifying interaction graphs is essentially unsupervised. That is, optimizing the unsupervised encoder is more challenging than the self-supervised decoder in nature. The key problem is how to match the learning speed between the interaction identification and the trajectory prediction so that both the tasks can be collaboratively optimized.

In this paper, we propose a novel framework named **O**nline **R**elational **I**nference (**ORI**) to efficiently identify hidden interaction graphs in evolving multi-agent interacting systems. Our method strategically allocates an adjacency matrix representing the interaction as a trainable parameter in the model and directly optimizes it through online backpropagation on the predicted trajectories. It effectively accelerates the update of the adjacency matrix than the encoder-based approach and ensures the following decoder to be learned with reasonable adjacency matrices from the early stage of training. ORI can seamlessly integrate with prior NRI models, offering architectural flexibility (*i.e.*, model-agnostic). Moreover, we developed an adaptive learning rate technique named AdaRelation particularly designed for relational inference in the evolving systems. It employs the historical adjacency matrix to indirectly estimate the decoder's sensitivity over the adjacency matrix and determine whether the learning needs to be accelerated. In addition, we introduce a data augmentation technique named Trajectory Mirror (TM) to expose various trajectories by flipping the axis in the systems. We experimentally demonstrate the effectiveness of ORI on various NRI models in both the synthetic and real-world (CMU MoCap for human motion [22]) datasets. Our key contributions are as follows:

- To the best of our knowledge, ORI is the first model-agnostic online relational inference framework for evolving multi-agent interacting systems. ORI employs online backpropagation to optimize an adjacency matrix from the trajectory information without any assumptions on the model architecture.

- We experimentally demonstrate that ORI identifies unknown interaction graphs in various evolving multi-agent interacting systems, such as sudden changes in interaction (Figure 2), parameters in the dynamics (Figure 3), and even dynamics itself (Figure 3), outperforming existing NRI models (Table 2).

- We propose AdaRelation, a novel adaptive learning rate technique particularly designed for relational inference in evolving multi-agent systems. It automatically tunes the learning rate for the adjacency matrix when interaction or/and dynamics in the system suddenly change (Figure 3).

- We propose Trajectory Mirror, a data augmentation technique to ensure the reasonable relational inference regardless of the trajectory axis. It significantly improves the convergence speed and overall interaction prediction accuracy in the several evolving scenarios (Supplementary).

## 2 Related Works

**Neural relational inference.** NRI [11] is the first work that formulated the problem of an unsupervised relational inference from observed agent trajectories. They implemented a VAE-based graph

neural network that encodes trajectories into an adjacency matrix, representing relation types, and then decodes the trajectory using the predicted adjacency matrix. This approach has been established as a standard in several follow-up works. For example, dNRI [12] and EvolveGraph [15] introduced graph recurrent network-based architectures to encode and generate dynamic priors for predicting evolving interactions. Additionally, a memory-augmented architecture has been proposed for enhanced long-term prediction using external associative memory pools [24]. More recently, finer granularities in relation modeling have been considered, such as edge to edge interaction [23], relation potentials using an energy-based function [25], and disentangled edge features [13]. However, all of these methods commonly rely on an encoder-decoder pair in an offline setting, supported by a large amount of batched training samples, without considering various evolving scenarios.

**Evolving multi-agent systems.**   Apart from relational inference, trajectory learning of evolving multi-agent interacting systems have actively explored using graph neural networks [18, 26, 6]. In particular, graph neural ordinary different equations (ODE) have been applied to various evolving scenarios, including evolving environments (*e.g.*, temperature and pressure) [18, 20] and stochastic motion [19]. However, these approaches assume a known graph structure, focusing on accelerating the simulation of physical dynamical systems [1, 7]. Additionally, each model addresses a specific type of evolution, raising questions about its generalizability to other scenarios. Moreover, adapting neural ODEs, potentially through online backpropagation, is a challenging problem since they are optimized from initial values [27].

**Online learning for multi-agent systems.**   Online learning for relational inference in multi-agent systems is a rarely explored research problem despite its significance. One primary related work [5] also treats the adjacency matrix as a trainable parameter without using an encoder, optimizing it with an online expert mixture algorithm, a type of online convex optimization algorithm. Consequently, the loss function needs to be convex over the adjacency matrix to guarantee optimization. Their model architecture is specifically designed to place message passing in the last layer of the model to ensure the model output is a linear combination of hidden states and the adjacency matrix, maintaining the convex property. However, this simple architecture significantly differs from recent architectures (*e.g.*, NRI and its variants), making it challenging to model nonlinear and complex interactions. Additionally, this work only reports trajectory prediction errors without providing any accuracy metrics on relational inference. Please note that a comparison with ORI is provided in the supplementary material.

## 3   Proposed Approach

### 3.1   Background

**Neural relational inference.**   Relational inference aims to identify an unknown interaction graph represented by an adjacency matrix $I^* \in R^{N \times N \times m}$, where $N$ is the number of agents and $m$ is the number of interaction types, from their trajectory in a given time period $x_{t:t+\Delta t}$, which generally include positions and velocities of agents. Existing approaches are mostly designed with a pair of neural network-based encoder and decoder, where encoder ($g_\phi$) predicts the adjacency matrix $I_t$ from an observed trajectory (*i.e.*, $g_\phi(x_{t-\Delta t:t})$ and then decoder predicts the future trajectory ($\hat{x}_{t:t+\Delta t}$) by $f_\theta(x_{t-\Delta t:t}, I_t)$. That is, the encoder is optimized to return the adjacency matrix for a lower error on the predicted trajectory (decoder output), assuming $I^* = \arg\min_{I_t} L(x_{t:\Delta t}, f_\theta(x_{t-\Delta t:t}, I_t))$.

**Online learning.**   Online learning problems have traditionally been formulated within the online convex optimization (OCO) framework, as introduced in [28]. Conventional gradient descent-based online optimization projects the updated variables onto a convex set at each step [29]. Such learning settings provide theoretical convergence guarantees when the loss function is convex with respect to the optimization variables. However, applying online gradient descent to deep neural networks (DNNs) is challenging due to the nonconvex nature of the loss function, and standard backpropagation performs poorly in an online setting [21]. Two primary directions of study have emerged to tackle the challenges of online backpropagation. One approach involves employing a flexible network structure, where the DNN architecture evolves over time [21, 30]. The other approach focuses on using an adaptive learning rate, where the learning rate is adjusted over time [31, 32]. Since our primary focus

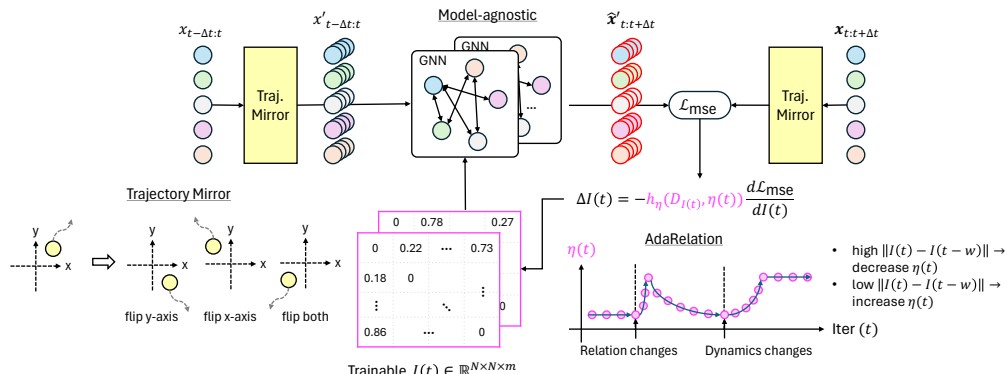

Figure 1: A brief illustration of the proposed Online Relational Inference (ORI) framework.

is on learning multi-agent interactions online without relying on any specific network structure, we concentrate on adapting the learning rate based on the evolution of the adjacency matrix.

## 3.2 Motivation of ORI

Our objective is to discover hidden relations between agents in evolving multi-agent interacting systems using their streaming trajectories. The motivation of this work comes from the primary challenges to apply existing methods to implement a fast-adapting, accurate, and stable online relational inference framework. Accordingly, we summarize our key motivations into three categories.

**Why we consider adjacency matrix as trainable parameter?** It may not be effective to simply apply the encoder and decoder-based existing methods to evolving multi-agent systems in the online setting. The primary challenge is that the intricate encoder is slowly trained with streaming and evolving data. It eventually degrades the decoder as well since the encoder and decoder are jointly trained, influencing each other. Whereas, ORI allocates the adjacency matrix as a trainable parameter, significantly enhancing the training speed in both the matrix and decoder. While such allocation is motivated from [5], this work still faces following crucial challenges.

**Why we need model-agnostic learning?** The relation inference is performed through the trajectory prediction without explicit supervision on graph structures (*i.e.*, true relation is not observable). This means the only supervision is defined by the predicted trajectories from the decoder, and hence largely depending on how effectively the decoder responds to changes in the embedding adjacency matrix. If a learning method is constrained to a specific decoder design, like [5], the performance achievable by the method can be constrained by the decoder design. Ideally, the learning method should offer the flexibility to seamlessly integrate to various decoder designs.

**Why we need adaptive learning rate?** The choice of a learning rate is particularly important in the evolving systems since the loss landscape can significantly vary with the evolution in the systems. For example, a low learning rate may be suitable for a slowly evolving dynamics A, but may a high one is needed for stable operation in a fast evolving dynamics B. A constant (or decaying) learning rate leads to a slow convergence or/and potentially a sub-optimal performance as the dynamics evolves over time. Ideally, the learning rate needs to be automatically tuned over time, avoiding a trade-off between faster convergence and unstable learning.

## 3.3 Training Procedures of ORI

First, the key difference in our training setup compared to the offline learning setup is that both batch and epoch are 1 (*i.e.*, streaming data). There is no separate validation or test dataset in online learning. Training of ORI is performed by online backpropagation on each individual training sample every iteration. Figure 1 describes the proposed approach. Note that the input and output to the decoder in the figure (denoted as GNN) and its optimization are essentially the same as existing methods. Our

contribution is primarily in the optimization method for the adjacency matrix given by:

$$I(t+1) = I(t) - h_\eta(D_{I(t)}, \eta(t))\frac{dL_{\mathrm{mse}}(\hat{x}_{t-\Delta t+1:t+\Delta t})}{dI(t)} \tag{1}$$

where $h_\eta(D_{I(t)}, \eta(t)) \in R^1$ is the relation-aware adaptive learning rate, elaborated in the following paragraph. Initially, the adjacency matrix ($I(t) \in R^{N \times N \times m}$; $I_{i,j}(t) \in [0,1]$) is filled with 0.5. Each training sample given to the model involves a single trajectory for a time period of $t - \Delta t : t + \Delta t$ (*i.e.*, $x_{t-\Delta t:t+\Delta t}$). The decoder ($f_\theta$) observes $x_{t-\Delta t:t}$ and then predicts the future trajectory by $\hat{x}_{t:t+\Delta t} = f_\theta(x_{t-\Delta t:t}, I(t))$ in an autoregressive manner. It also reconstructs the trajectory $\hat{x}_{t-\Delta t+1:t}$ during the observation. ORI estimates a MSE loss on the predicted and reconstructed trajectory at $t + \Delta t$ (*i.e.*, $L_{\mathrm{mse}}(\hat{x}_{t-\Delta t+1:t+\Delta t})$) and then updates the decoder and adjacency matrix using gradient descent. Note that, the adjacency matrix is updated only once at each iteration. The model infers the relation in the given trajectory at the last time step. While the training objective is to minimize the overall MSE loss throughout the streaming data, our primary evaluation criterion is the relation accuracy representing the proportion of true positive and true negative in the adjacency matrix.

**Adaptive relation-aware learning rate.** We propose AdaRelation, a learning rate adaptation technique specifically designed for relational inference. AdaRelation adjusts the learning rate for the adjacency matrix (not the decoder) based on changes in the norm of the adjacency matrix. For example, if the norm exceeds a certain threshold, it gradually decreases the learning rate within a defined range.

The mechanism is intuitive: a good trajectory predictor should show a large output variance depending on the adjacency matrix, meaning the quality of a predicted trajectory should vary clearly with changes in the adjacency matrix. We observe that the norm of gradient ($||\frac{dL_{\mathrm{mse}}}{dI(t)}||_1$) indeed increases as the model adapts to the evolved system, often making the adjacency matrix unstable (see (1)). Conversely, it decreases when the system suddenly evolves, slowing down the update of the adjacency matrix. Therefore, the gradient norm indicates when the update of the adjacency matrix needs to be stabilized or accelerated. Given the adjacency matrix is the sum of this gradient over time, comparing the current matrix with a past one essentially reflects changes in the gradient norm. Accordingly, we define $D_{I(t)}$ to estimate the evolution in the L1 norm of the adjacency matrix over $w$ time steps as follows:

$$D_{I(t)} = \frac{1}{N^2 m}||I(t) - I(t-w)||_1 \tag{2}$$

This measures how much the predicted interaction strength ($I_{i,j,k}$) changes on average over $w$ time steps. We define a threshold parameter $\epsilon$ to restrict changes in $I_{i,j,k}$ to be remain near this threshold. The update of $\eta(t)$ involves adding or subtracting $\alpha$, an adaptation step size, determined by the range of $D_{I(t)}$ and the threshold:

$$\Delta\eta(t) = \begin{cases} -\alpha & \text{if } D_{I(t)} > \epsilon \\ \alpha & \text{otherwise} \end{cases} \tag{3}$$

Consequently, the next learning rate ($\eta(t+1)$) is represented by $h_\eta(D_{I(t)}, \eta(t))$:

$$h_\eta(D_{I(t)}, \eta(t)) = \mathrm{clip}_{\eta_{\min};\eta_{\max}}(\eta(t) + \Delta\eta(t))) \tag{4}$$

where $\mathrm{clip}_{\eta_{\min};\max}$ limits the learning rate within the lower bound ($\eta_{\min}$) and upper bound ($\eta_{\max}$). This effectively controls the learning rate to automatically stabilize and accelerate the update of the adjacency matrix. More intuition behind AdaRelation is discussed in the supplementary material.

**Trajectory Mirror.** The models trained by the online backpropagation are often prone to be biased to certain training samples. It also happens in multi-agent interacting systems. For example, the coordinates and velocities of agents in the currently observed data samples can be biased. Such scenario has not been discussed much in literature because existing works expose the model to a huge amount of simulations with short-term trajectories, ensuring several different initial positions and velocities. However, our problem addresses streaming trajectories in a much longer-term, where we do not have an access to initialize their positions and velocities. Ideally, the relation between agents should be correctly inferred, regardless where the model observes them. Accordingly, we consider a data augmentation technique named Trajectory Mirror, which flips the axis and generates,

for example in a two-dimensional space, three additional trajectories (see Figure 1). It is simple yet effective to avoid the model bias and significantly enhance the convergence speed of the model without storing the past trajectories. Ablation studies are available in the supplementary material.

**Technical novelty.** ORI is a novel model-agnostic online relational inference framework, incorporating a strategic combination of two different learning methods for the adjacency matrix and decoder. ORI introduces AdaRelation, an adaptive learning rate technique designed for relational inference, and Trajectory Mirror, a simply yet effective data augmentation technique proved in several evolving scenarios (results are in supplementary). Our approach clearly differs from existing methods which perform end-to-end gradient descent on the encoder [11, 12, 23] or online convex optimization constrained to a specific decoder design [5].

## 4 Experimental Results

**Dataset.** We experimented with three common benchmarks, synthetic springs and charged systems, and CMU MoCap (human motion) [22]. The synthetic datasets were generated using the open-source code from NRI [11]. For the springs and charged systems, 10 simulations with 90k time steps each were created, involving 10 agents with different random interactions ($p = 0.5$). These simulations are sequentially presented to models. The evolving relation dataset features different interaction graphs in each simulation with fixed spring $k = 0.1$ and $k_e = 1.0$. Another dataset varies these constants, generated from uniform distributions $[0.1, 0.2]$ for springs and $[1, 2]$ for charged constants. The evolving relation and dynamics dataset randomly selects each simulation from either the springs or charged systems. The CMU MoCap dataset, processed using dNRI's open-source code [12], includes human motion data with 31 joints and various actions, captured at 120Hz.

**Baseline and implementation detail.** ORI is a model-agnostic online learning framework involving the trainable adjacency matrix. The adjacency matrix is initialized only at the initialization stage. The same adjacency matrix is used throughout the entire samples and simulations without assuming that we know when the interaction evolves. ORI can integrate with various models as long as they use the adjacency matrix as input. Most prior works use decoders based on graph multi-layer perceptron (MLP) and graph recurrent neural networks (RNN), as seen in NRI [11], incorporating node-to-edge and edge-to-node message passing. Additionally, MPM [23] introduces a graph RNN and an attention-based graph RNN for more efficient edge-to-edge message passing. To demonstrate ORI's effectiveness, we use four different trajectory predictors: NRIm (MLP-based), NRIr (RNN-based), MPMr (RNN-based), and MPMa (attention-based). Since ORI employs the decoders in NRI and MPM, we primarily compare our approach with the original NRI and MPM. We also include dNRI [12], which is specifically designed for dynamically evolving interaction graphs. As these works analyzed the performance in the offline setup, we evaluated their models in the online setup. We follow their default implementation but replace the encoder with the trainable adjacency matrix. More details on the training setup are available in the supplementary material.

### 4.1 Inferring Relation in Evolving Interaction Graph

We explore relation inference accuracy in the synthetic systems incorporating evolving interaction graphs. These systems involve constant parameters and no switching in the dynamics. Figure 2 demonstrates the relation accuracy over time and predicted trajectories in the springs system. The relation accuracy is evaluated based on the number of true positives and true negatives excluding self-interaction (*i.e.*, total $10 \times 9$ relations). Our approach is able to quickly recover the relation accuracy when the model encounters a new interaction graph. For example, at the bottom of Figure 2(a), the target interaction matrix suddenly changes at the 15k-th iteration. While it fails to adapt in a single iteration, the target and predicted matrices are aligned well after 100 iterations. In contrast, the baseline (MPM) accuracy slowly increases throughout the entire training iterations, showing a significant gap in the average relation accuracy. Figure 2(b) compares the target and predicted trajectories at the 15k-th iteration (*i.e.*, right after the new interaction) and 18k-1-th iteration (*i.e.*, right before another interaction), indicating the trajectory quality also improves along with the accuracy.

Table 2 showcases the average accuracy and average MSE loss during entire iterations in the springs and charged systems. As the interaction graph changes over time, simply reporting the final accuracy only represents how well the model adapts to the last graph. Accordingly, we report the average

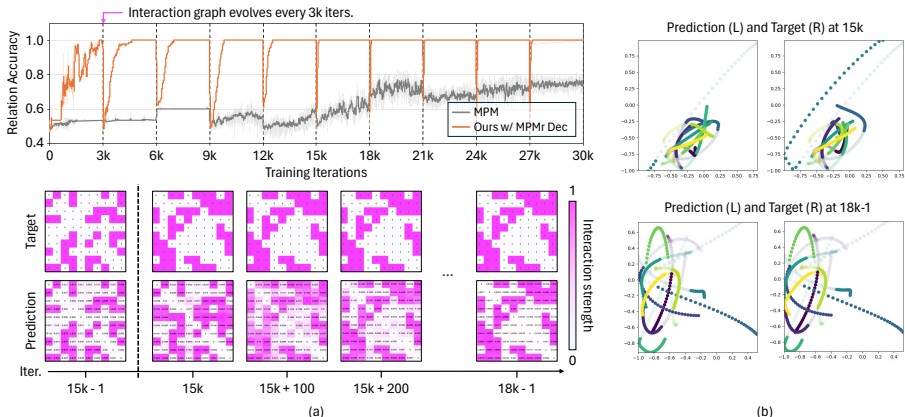

Figure 2: Prediction results of ORI with MPMr decoder and the baseline MPM in the springs system. (a) the relation accuracy in the two models throughout the training (top) and visualization of the target and predicted adjacency matrix in our model (bottom). (b) target and predicted trajectories in our model.

Table 2: Comparison with offline learning models in springs and charged systems with evolving interactions. Acc and mse stand for the relation accuracy and mse on the predicted trajectory averaged over the entire training iterations. The number following mse (*e.g.*, mse 10) denotes the mse at the 10-th prediction time step.

| Method | Springs | | | | | Charged | | | | |
|---|---|---|---|---|---|---|---|---|---|---|
| | Acc (%) | mse 1 | mse 10 | mse 20 | mse 30 | Acc (%) | mse 1 | mse 10 | mse 20 | mse 30 |
| dNRI [12] | 51.1 | 4.34e-5 | 8.07e-4 | 2.40e-3 | 5.34e-3 | 50.3 | 2.02e-3 | 5.00e-3 | 9.44e-3 | 5.40e-2 |
| NRI [11] | 57.4 | 1.70e-4 | 1.36e-3 | 4.49e-3 | 1.11e-2 | 52.0 | 3.80e-3 | 1.13e-2 | 2.40e-2 | 4.63e-2 |
| MPM [23] | 61.6 | 2.76e-4 | 1.10e-3 | 4.10e-3 | 9.15e-3 | 51.7 | 6.16e-3 | 1.21e-2 | 2.60e-2 | 4.86e-2 |
| Ours w/ NRIm | 74.5 | 1.45e-4 | 8.19e-4 | 2.71e-3 | 7.32e-3 | 88.6 | 6.60e-3 | 1.61e-2 | 3.79e-2 | 7.33e-2 |
| Ours w/ NRIr | 94.2 | 1.24e-4 | 6.02e-4 | 1.71e-3 | 3.44e-3 | 90.9 | 4.87e-3 | 1.47e-2 | 3.54e-2 | 6.85e-2 |
| Ours w/ MPMr | 95.2 | 2.80e-4 | 4.54e-4 | 1.43e-3 | 3.16e-3 | 95.3 | 6.98e-3 | 1.35e-2 | 3.06e-2 | 5.91e-2 |
| Ours w/ MPMa | 96.4 | 2.59e-4 | 3.74e-4 | 1.17e-3 | 2.62e-3 | 87.1 | 6.85e-3 | 1.42e-2 | 3.25e-2 | 6.26e-2 |

accuracy over entire iterations to understand the model's accuracy on the multiple graphs and how fast it adapts to the change in the graphs. ORI with four different decoders consistently outperform the existing encoder and decoder-based methods with respect to the accuracy. Note that TM is applied to both existing methods and ours for the fair comparison. It is crucial to emphasize that, in the charged system, our results with almost 40% higher accuracy does not exhibit lower trajectory errors. For example, NRI reaching an accuracy of 52.0% still shows the lower MSE of $3.80 \times 10^{-3} \sim 4.63 \times 10^{-2}$ over all the prediction steps than our results. This implies the comparison solely using the MSE loss may not indicate the quality of inferred relations at all. The lower MSE in existing methods is probably achieved by the larger capacity than our methods due to the encoder, overfitting to the trajectory modeling, not the relation inference. However, in spring systems, the methods with higher accuracy exhibits lower trajectory errors.

## 4.2 Inferring Relation in Evolving Interaction Graph and Dynamics

ORI is evaluated on the two additional evolving scenarios in the springs (spr) and charged (cha) systems, where 1) interaction graph and parameter evolve and 2) interaction graph and the dynamics itself evolve. In addition, we analyze the relation learning rate to understand how the relation accuracy responds depending on the learning rate. The yellow, red, and blue curves correspond to ORI with AdaRelation and ORI with constant learning rates (lower and upper bound). Figure 3 presents the relation accuracy and relation learning rate over 30k training iterations in both the evolution scenarios. Although the models tend to slowly converge compared to the springs systems, they can still adapt to the systems with evolving parameters or switching dynamics, reaching the 1.0 accuracy given enough training iterations.

An interesting observation lies in changes of the relation learning rate. For example, in Figure 3 (left bottom), the learning rate in AdaRelation (yellow) automatically increases at the 12k-th

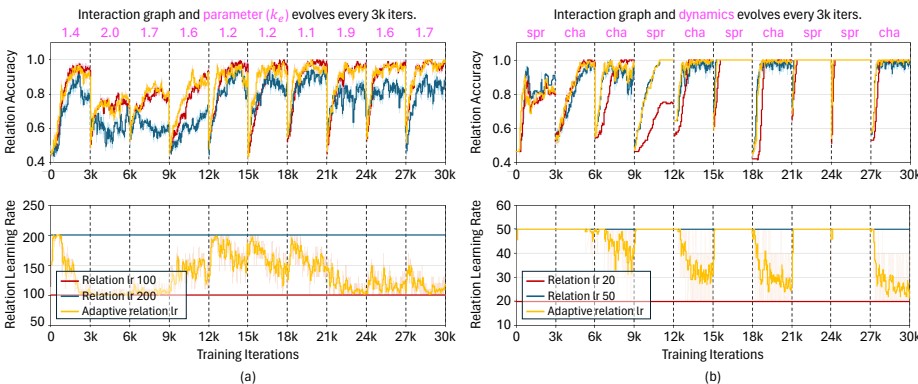

Figure 3: Prediction results of ORI with NRIr decoder in the charged system with evolving interaction and parameters (a) and ORI with MPMr decoder in the springs and charged systems with evolving interaction and dynamics (b). 1-st row compares the relation accuracy between constant learning rates and AdaRelation. 2-nd row shows changes in the relation learning rate throughout the training.

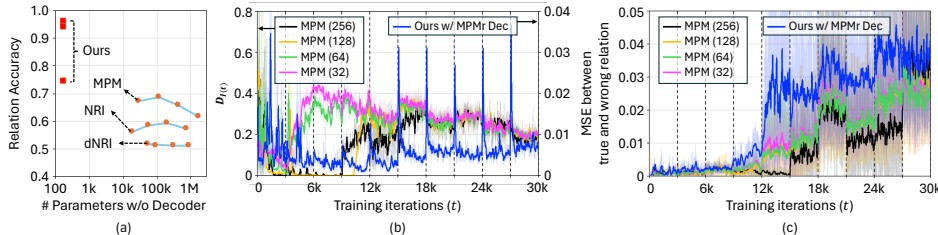

Figure 4: Comparison between ORI and existing methods with respect to the relation accuracy (a), variance in the adjacency matrix (b), and variance in the predicted trajectory (c) depending on encoder complexity. The number in the MPM (·) represents the dimension of hidden states in the encoder.

iteration, decreases over time, and then increases again at the 15k-th iteration. This means that AdaRelation notices a change in the interaction graph of the systems in an unsupervised manner and hence increases the learning rate for a while. This ensures not only the fast adaptation to a new environment but also the stability (*i.e.*, less fluctuation in the accuracy) after the relation accuracy converges. Moreover, Figure 3 (right bottom) demonstrates that AdaRelation controls the learning rate depending on the dynamics as well. The models with a high learning rate are stable enough in the springs systems. However, the fluctuation in the accuracy emerges in the charged systems, particularly when the accuracy almost reaches 1. AdaRelation effectively suppresses such fluctuation without sacrificing the convergence speed. For example, between the 18k-th and 21k-th iterations, the accuracy of AdaRelation converges as fast as the high learning rate's one while having much less fluctuation in the later iterations. Thus, AdaRelation effectively enhances the convergence speed, stability, and overall accuracy in evolving multi-agent interacting systems. The related accuracy and ablation studies are available in the supplementary material.

## 4.3 Discussion on Performance Gain in ORI

We discuss how existing methods fail to clarify the benefit of ORI over them. Since they share the same trajectory predictors, the primary reason should be studied with respect to the encoder side.

**Lightweight encoders in existing works.** One of the key features in ORI is the encoder-less design, having much fewer trainable parameters, excluding the trajectory predictor, than existing works. To clarify whether the performance gain is from the less trainable parameters, we demonstrate how existing methods perform when their encoder is significantly lightened while having the same decoder. Figure 4(a) showcases the relation accuracy depending on the encoder complexity in the spring system with evolving interaction. However, the performance gain from their lightweight encoders is not significant, indeed still much lower than ORI. Accordingly, we explore the following two additional experiments.

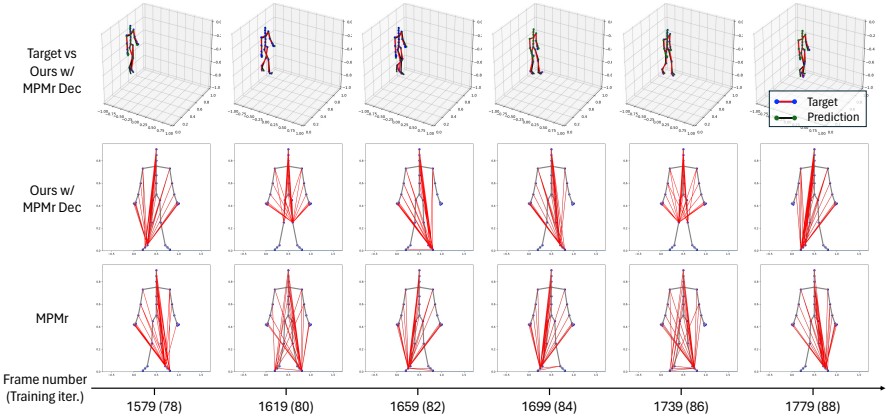

Figure 5: Prediction results of ORI with MPMr decoder and MPM in CMU MoCap dataset. 1-st row represents the last frame in the predicted and target trajectory from ORI. 2-nd and 3-rd rows visualize the top-30 stongest interaction edges in the corresponding frame from ORI and MPM. Note that MPM allocate higher relation strengths in the front foot while ORI focuses on the foot behind.

**Changes in inferred relations.** Figure 4(b) describes changes in the adjacency matrix ($D_{I(t)}$) throughout training in the models with different encoder complexity. Note that $D_{I(t)}$ estimates how much the current adjacency matrix differs from the past one (*i.e.*, $\sim ||I(t) - I(t - w)||_1$). First, the range of $D_{I(t)}$ in ours is consistent throughout the entire iterations, effectively updating the adjacency matrix from the early stage of training. In contrast, MPMs even with the smaller encoder, such as MPM (64) and MPM (32), exhibit sudden increases in $D_{I(t)}$ after few thousands iterations. This means that the predicted adjacency matrix is not responding much to the observed trajectory, implying their encoders completely fail to discover the relationship between the observed trajectory and the adjacency matrix. The larger models, MPM (256) and MPM (128), show worse results such as slow increases in $D_{I(t)}$ and almost no changes in the first 10k training iterations. That is, the lightweight encoder still shows the totally different behavior to ORI and does not effectively update the adjacency matrix in the early stage.

**Output variance given true and wrong relations.** Following the above paragraph, we study how the slow optimization of the encoder influences the trajectory predictor. Figure 4(c) displays the gap in MSE losses between the output with true interaction graph and one with completely wrong interaction graph over the training iterations. This essentially represents how sensitive the trajectory predictor is to the input interaction graph. ORI demonstrates a clearly larger gap in the MSE losses than all the other MPM models. Apart from ORI, the smaller encoders increase the MSE gap in the trajectory predictor. That is, the failure in the encoder degrades the trajectory predictor, which potentially influences the encoder again. In summary, allocating the trainable adjacency matrix in ORI ensures the stable update in the embedding adjacency matrix to the trajectory predictor and enhances its output variance depending on the interaction graph.

## 4.4 Real-world Application

ORI is assessed in the real-world human motion dataset (CMU MoCap) [22] against existing offline methods. Figure 5 showcases the target and predicted trajectories on walking motion from ORI (1-st row) and the top-30 strongest relations between joints in a skeleton model for the corresponding frame (2-nd row). Additionally, the figure incorporates the inferred relations from MPM (3-rd row). The visual comparison illustrates that ORI's predicted joint trajectories closely align with the target, yet ORI exhibits higher MSE loss compared to MPM (see the supplementary material). However, similar with the observation in the charged systems (Table 2), ORI appears to offer more interpretable relation inference on the joints. For example, in 3-rd row of the figure, MPM simplifies the relational inference by cyclically focusing on right foot, left foot, and right foot again. In contrast, ORI introduces an additional layer of shifts in the relation, emphasizing primary connections between left foot, right knee, and right foot. This additional detail improves the interpretability of the walking motion.

# 5 Conclusion

**Summary.** We introduced Online Relational Inference (ORI), a novel framework for online relational inference in evolving multi-agent interacting systems. ORI employs an adaptive learning rate technique, AdaRelation, allowing it to adapt dynamically to changing environments through online backpropagation. Our approach also includes the Trajectory Mirror (TM) data augmentation method to enhance generalization. This model-agnostic framework seamlessly integrates with various neural relational inference architectures, providing a robust solution for real-time applications in complex, evolving systems. Future work will focus on enhancing the adaptability in more fast-evolving interaction and exploring its applicability to a wider range of multi-agent systems.

**Limitation and future work.** Our current experiments do not evaluate ORI in non-ideal environments, incorporating directed interaction graphs or/and variable number of nodes. This limits the potential of ORI in relatively ideal environments. We expect that ORI can be extended to scenarios when agents are added or deleted, provided we are aware of which agent is added and deleted by adding and deleting corresponding row and column in the adjacency matrix. While the experiments in the paper do not explicitly include directed interactions, ORI does not assume any symmetry (undirected graph) in the adjacency matrix. In other words, there is no technical limitation to apply ORI in directed interaction.

## Acknowledgements

This work is partly supported by the Office of Naval Research under Grant Number N00014-20-1-2432 and National Science Foundation under Grant Number 2328962. The views and conclusions contained in this document are those of the authors and should not be interpreted as representing the official policies, either expressed or implied, of the Office of Naval Research, National Science Foundation, or the U.S. Government.

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

# A    Training Setup

**Training setup.** For the synthetic datasets, the model observed the first 30 time steps ($x_{t-30:t}$) and then predict the next 30 time steps ($x_{t:t+30}$). The next prediction window is defined on $x_{t+30:t+90}$. Similarly, for the CMU MoCap data, the model observed the first 10 time steps and then predict the next 10 time steps. We trained the models with a single GTX 2080Ti GPU.

**Implementation details.** We directly use the implemented decoders from NRI [11] and MPM [23] for ORI with NRI and ORI with MPM. The hidden size on the decoders are set to 256 as default. The learning rate for the decoders is 1e-4.

The initial learning rates ($\eta(0)$) for the adjacency matrix is 100 for ORI with NRI decoders and 20 for ORI with MPM decoders. The lower and upper bound for learning rates in AdaRelation are $\eta_{\min} = 100$ $\eta_{\max} = 200$ for ORI with NRI decoders, $\eta_{\min} = 20$ $\eta_{\max} = 50$ for ORI with MPM decoders. The threshold $\epsilon$ and adaptation step size $\alpha$ in AdaRelation are set to 0.05 and 1. For the CMU MoCap dataset, we observe that the gradient on the adjacency matrix is relative small; in order to ensure more dynamic updates in the adjacent matrix, we largely increase the $\eta(0)$ to 100k.

# B    Additional Discussion

**Computational complexity.** In terms of trainable parameters, NRI has 721.4k for encoder and 727.3k for decoder; MPM has 1724.9k for encoder and 1071.7k for decoder; dNRI has 883.7k for encoder and 269.8k for decoder. In terms of FLOPs per iteration, NRI shows 177.8MFLOPs for encoder and 5040.5MFLOPs for decoder; MPM shows 3.9GFLOPs for encoder and 10.9GFLOPs for decoder. It indicates that the decoder consumes more computation than the encoder even though they are with the similar level of trainable parameters.

**Intuition behind the norm of gradient** $||\frac{dL_{\mathrm{mse}}}{dI(t)}||_1$. The norm of gradient indicates that how the trajectory error ($\Delta L_{\mathrm{mse}}$) changes by the adjacency matrix ($\Delta I(t)$). Ideally, we expect this norm being high enough so that the model learns the strong correlation between the trajectory of agents and their relation. In other words, the low norm of gradient means that, the model returns the similar trajectory regardless of the relation (*i.e.*, adjacency matrix), which is undesirable.

**Intuition behind the deviation** $D_{I(t)}$. The deviation is a function of $||I(t)-I(t-w)||_1$, where both $I(t)$ and $I(t-w)$ are the learned adjacency matrix, not the actual one. Hence, the significant change in the actual adjacency matrix used for generating the observed time series, may not necessarily lead to large values of the deviation. This depends on how quickly ORI learns the new adjacency matrix as discussed below.

Consider a scenario in which the actual adjacency matrix significantly changes between the time steps $t - w$ and $t$. Assume ORI has learned the actual adjacency matrix in the time step $t - w$. Now, in the time step $t$, ORI can respond in two possible ways.

First, ORI may quickly identify the new adjacency matrix at the time step $t$. In this case, $I(t)$ and $I(t - w)$ will be related to the new and previous actual adjacency matrices, respectively. Assuming the two actual adjacency matrices are significantly different, the deviation between two learned adjacency matrices will also be large. Hence, based on equation (3), the learning rate will decrease. This will make the learned adjacency matrix stable, thereby helping ORI to stay at the new learned adjacency matrix at the time $t$, which is desirable as that is also the actual adjacency matrix.

Let us now consider the second case where ORI does not quickly learn the new actual adjacency matrix and hence, the learned adjacency matrix at time $t$ stays close to the one learned at time step $t - w$. In other words, $||I(t) - I(t - w)||_1$ remains low even if the actual adjacency matrices have changed. In this case, following equation (2), the learning rate increases to rapidly update the learned adjacency matrix, which is desirable to quickly move the learned matrix to the actual one.

In summary, the equation (2) and (3) appropriately update the learning rate when actual adjacency matrix changes, even without any knowledge/supervision of that actual matrices.

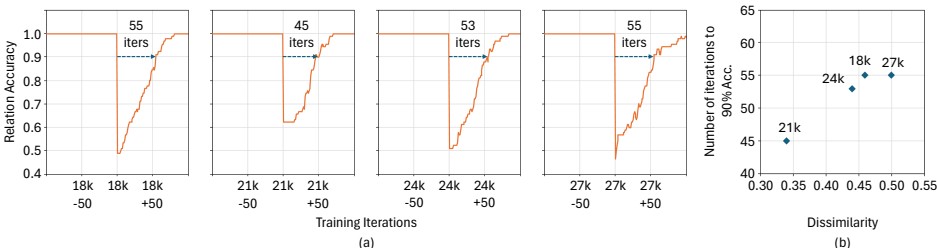

Figure 6: Correlation between the dissimilarity and the number of iterations required to reach 90% accuracy since the interaction graph evolves in ORI with MPMr decoder in the springs system.

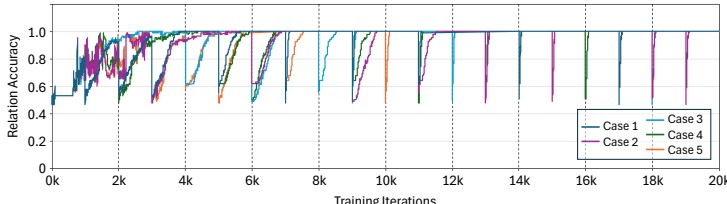

Figure 7: ORI with MPMr decoder in five different cases with irregular evolution in interaction. The system is based on springs system with 10 agents and consists of three 1k iterations, four 2k iterations, and three 3k iterations.

## C  Additional Experimental Results

**Correlation between dissimilarity in graph and relational accuracy.**    Even though we randomly evolved the interaction graphs, there exists a slight similarity between the graphs. A dissimilarity is defined on two interaction graphs to understand how it influences relational accuracy. We first sum the element-wise difference between two interaction graphs and then divide by the number of elements. This dissimilarity is compared with the number of iterations required to reach 90% accuracy since the interaction graph evolves. For example, in ORI with MPMr decoder in the springs system (Figure 2 in the main paper), in the early stage of training, we do not observe much correlation between the dissimilarity, as the model is not yet matured. However, after approximately 18k iterations, more dissimilar interaction graphs require more training iterations to reach the accuracy 90% (see Figure 6). Note that, higher dissimilar interaction graphs indicate the higher accuracy drop when the interaction graph evolves. This leads to the lower initial accuracy and hence requires less iterations. (see the second column in Figure 6(a)). However, we consider the variation of the required training iterations is still marginal (*e.g.*, 45-55 iterations in Figure 6(b)).

**Irregular evolution in relations.**    We consider five different cases with irregular evolutions in interaction as follows. The relational accuracy over iterations is described in Figure 7.

Case 1: interaction graph changes after 1k, 1k, 1k, 2k, 2k, 2k, 2k, 3k, 3k, 3k iterations.

Case 2: interaction graph changes after 3k, 3k, 3k, 2k, 2k, 2k, 2k, 1k, 1k, 1k iterations.

Case 3: interaction graph changes after 1k, 3k, 2k, 2k, 1k, 3k, 2k, 3k, 1k, 2k iterations.

Case 4: interaction graph changes after 2k, 3k, 1k, 3k, 2k, 2k, 1k, 2k, 3k, 1k iterations.

Case 5: interaction graph changes after 3k, 1k, 1k, 2k, 3k, 2k, 2k, 1k, 3k, 2k iterations.

From Case 1 to Case 5, the accuracy is 93.9%, 92.2%, 93.8%, 93.1%, and 92.5%. Overall, the variation in the accuracy is marginal, and hence, the performance of ORI is not significantly influenced by the irregular evolutions. However, since we only considered 1k, 2k, and 3k iterations, more extreme scenarios, such as few-thousands iterations would be an interesting future study.

**MSE in CMU MoCap.**    Table 3 summarizes the MSE loss on the predicted trajectory in the CMU MoCap dataset. Overall, the difference in the MSE losses are not significant. However, the original MPM method works better than ORI with NRI and ORI with MPM.

Table 3: Comparison of MSE loss on the predicted trajectory in the CMU MoCap dataset.

| Method | CMU MoCap | | |
|---|---|---|---|
| | mse 1 | mse 5 | mse 10 |
| NRI [11] | 1.23e-4 | 6.64e-3 | 2.32e-3 |
| MPM [23] | 1.17e-4 | 5.28e-3 | 1.49e-3 |
| Ours w/ NRIr | 1.12e-4 | 6.57e-4 | 1.86e-3 |
| Ours w/ MPMr | 1.31e-4 | 7.23e-4 | 2.22e-3 |

Table 4: Comparison with prior online learning method in springs systems with 20 agents.

| Method | Relative MSE compared to NRI [11] | | | | | Relative MSE compared to dNRI [12] | | | | |
|---|---|---|---|---|---|---|---|---|---|---|
| Prediction step | 1 | 2 | 5 | 8 | 10 | 1 | 2 | 5 | 8 | 10 |
| CoGNN-AR [5] | 0.339 | 0.345 | 0.617 | 0.680 | 0.635 | 0.731 | 0.714 | 1.194 | 1.373 | 1.312 |
| CoGNN-LSTM [5] | 0.268 | 0.259 | 0.317 | 0.340 | 0.371 | 0.577 | 0.536 | 0.613 | 0.686 | 0.766 |
| CoGNN-TC [5] | 0.286 | 0.310 | 0.300 | 0.340 | 0.352 | 0.615 | 0.643 | 0.581 | 0.686 | 0.727 |
| Ours w/ NRIm | 0.565 | 1.655 | 0.672 | 0.578 | 0.563 | 0.565 | 0.631 | 0.863 | 1.048 | 1.151 |
| Ours w/ MPMr | 1.028 | 1.681 | 0.359 | 0.253 | 0.228 | 1.028 | 0.641 | 0.461 | 0.459 | 0.466 |
| Ours w/ MPMa | 1.653 | 1.610 | 0.279 | 0.181 | 0.165 | 1.653 | 0.614 | 0.359 | 0.329 | 0.337 |

**Comparison with prior online work.** The primary related work for online learning in multi-agent interacting systems is [5]. They also considered springs systems with 20 agents, but their code, implementation details, and relation accuracy are not available, making direct comparison difficult. Instead, we provide an indirect comparison using similar data (springs system with 20 agents and evolving interaction). This work compared their method with NRI [11] and dNRI, so we normalize their results with their NRI and dNRI, and ours with our NRI and dNRI, and then compare these results as relative MSE (Table 4). We observe that in the very early prediction steps (*e.g.*, steps 1 and 2), their method performs better, but ours achieves a clearly lower relative MSE loss in the later time steps.

**Ablation study.** We perform ablation studies on AdaRelation and Trajectory Mirror in the three different evolving systems. For AdaRelation, the models with constant learning rates, lower or upper-bound learning rate in AdaRelation, are compared. Table 5, Table 6, and Table 7 summarize the results. We bold the highest accuracy (or others with less than 0.5% difference). Overall, the large accuracy gain is resulted from Trajectory Mirror, and AdaRelation further enhances the accuracy.

Table 5: Ablation study in springs and charged systems with evolving interaction.

| Method | Springs with evolving interaction | | | Charged with evolving interaction | | |
|---|---|---|---|---|---|---|
| | Trajectory Mirror | AdaRelation | Acc (%) | Trajectory Mirror | AdaRelation | Acc (%) |
| Ours w/ NRIr | ✓ | constant (100) | 90.2 | ✓ | constant (100) | 89.3 |
| Ours w/ NRIr | ✓ | constant (200) | **94.4** | ✓ | constant (200) | 87.0 |
| Ours w/ NRIr | ✓ | ✓ | **94.2** | ✓ | ✓ | **90.9** |
| Ours w/ MPMr | | ✓ | 80.5 | | ✓ | 90.2 |
| Ours w/ MPMr | ✓ | constant (20) | 93.1 | ✓ | constant (20) | 92.2 |
| Ours w/ MPMr | ✓ | constant (50) | **95.5** | ✓ | constant (50) | 94.1 |
| Ours w/ MPMr | ✓ | ✓ | **95.2** | ✓ | ✓ | **95.3** |

Table 6: Ablation study in springs and charged systems with evolving interaction and parameter.

| Method | Springs with evolving interaction and parameter | | | Charged with evolving interaction and parameter | | |
|---|---|---|---|---|---|---|
| | Trajectory Mirror | AdaRelation | Acc (%) | Trajectory Mirror | AdaRelation | Acc (%) |
| Ours w/ NRIr | ✓ | constant (100) | **94.9** | ✓ | constant (100) | 83.4 |
| Ours w/ NRIr | ✓ | constant (200) | **95.0** | ✓ | constant (200) | 73.8 |
| Ours w/ NRIr | ✓ | ✓ | **94.7** | ✓ | ✓ | **84.8** |
| Ours w/ MPMr | | ✓ | 85.1 | | ✓ | 79.6 |
| Ours w/ MPMr | ✓ | constant (20) | 93.4 | ✓ | constant (20) | **92.5** |
| Ours w/ MPMr | ✓ | constant (50) | **95.4** | ✓ | constant (50) | 88.8 |
| Ours w/ MPMr | ✓ | ✓ | **95.0** | ✓ | ✓ | **92.1** |

Table 7: Ablation study in springs and charged systems with evolving interaction and dynamics.

| Method | Evolving interaction and dynamics (springs and charged) | | |
|---|---|---|---|
| | Trajectory Mirror | AdaRelation | Acc (%) |
| Ours w/ MPMr | | ✓ | 54.5 |
| Ours w/ MPMr | ✓ | constant (20) | 86.5 |
| Ours w/ MPMr | ✓ | constant (50) | 90.7 |
| Ours w/ MPMr | ✓ | ✓ | **91.3** |

