# OpenReview forum: "Online Relational Inference for Evolving Multi-agent Interacting Systems"
_NeurIPS.cc/2024/Conference — NeurIPS 2024 poster_

### Official Review · Reviewer_arYB · 2024-06-20

**Soundness:** 3
**Presentation:** 2
**Contribution:** 2
**Rating:** 5
**Confidence:** 1

**Summary:**

The paper introduces the Online Relational Inference (ORI) framework to identify hidden interaction graphs in evolving multi-agent systems using streaming data. The framework employs online backpropagation, updating the model with each new data point, thus adapting to dynamic environments in real-time. ORI features a trainable adjacency matrix optimized through an adaptive learning rate technique, which adjusts based on the historical sensitivity of the decoder to changes in the interaction graph. Experimental results on synthetic datasets and the CMU MoCap dataset demonstrate ORI's effectiveness in improving relational inference accuracy and adaptability.

**Strengths:**

1. The paper is well-written.
2. The studied problem is interesting.
3. The experimental validation on both synthetic and real-world datasets is sufficient.

**Weaknesses:**

1. The efficiency analysis is not sufficient. I suggest authors should include the comparison of training time.
2. The compared baselines are weak and limited. Authors should include stronger baselines published in 2023-2024.
3. The technical contribution is a little weak. The Eqn. 1 seems to be a natural way to optimize the graph structure. Moroever, I'm not sure whether there is overfitting when minimizing Eqn. 1.
4. The limitation of the work is not sufficiently discussed.

**Questions:**

See above.

---

> ### Author Rebuttal · Authors · 2024-08-07
>
> **W1. [Efficiency Analysis]** Thank you for the suggestion. We agree that more efficiency analysis on ORI will provide important information to readers. We compare the overall computational complexity, including the number of trainable parameters, FLOPs, and running time, in ORI with NRIr decoder and ORI with MPMr decoder with the encoder-based methods (e.g., NRI, MPM, and dNRI).
> In terms of trainable parameters, NRI has 721.4k for encoder and 727.3k for decoder; MPM - 1724.9k for encoder and 1071.7k for decoder; dNRI - 883.7k for encoder and 269.8k for decoder. In terms of FLOPs per iteration, NRI shows 177.8MFLOPs for encoder and 5040.5MFLOPs for decoder; MPM – 3.9GFLOPs for encoder and 10.9GFLOPs for decoder. Note, FLOPs are estimated by fvcore package, but dNRI was not compatible. While the decoder and encoder have the similar level of trainable parameters (e.g., NRI and MPM), the decoder often shows much higher FLOPs due to the repeated trajectory prediction while the encoder predicts a single interaction graph per trajectory.
>
> For the running time in a single 2080Ti GTX GPU, NRI shows 0.194sec/iter; MPM – 0.183sec/iter; dNRI – 0.096sec/iter. ORI with NRIr decoder shows 0.235sec/iter; ORI with MPMr decoder – 0.192sec/iter. Although ORI shows slightly higher running time than the encoder-based models, theoretically, as ORI does not have an encoder network and apply the gradient descent to much smaller adjacency matrix, its FLOPs is approximated to the only decoder’s FLOPs. In addition, note that, their running time will be largely depending on the decoder design as the decoder is the major computational bottleneck.
> We observe that the latency in ORI occurs at the unoptimized code for two separate gradient descent (one for the decoder and another one for the adjacency matrix). We are working on optimizing the run-time of that step in ORI.
>
> **W2. [Recent Baselines]** The reviewer's concern is valid. We experimented another very recent baseline “GDP” [1] in springs and charged systems (same setup in Table 2 in main paper). GDP shows 68.9% accuracy in springs systems and 51.4% accuracy in charged systems. While the accuracy in the springs system is much higher than other encoder-based methods, it is still much lower than ORI-based methods (~96.4%). The performance gap is even higher in the changed system. Please see the result in Table 2 in the attached pdf. We consider exploring more recent baselines and applying ORI on top of these methods will be an interesting future work.
>
> [1] Pan, Liming, Cheng Shi, and Ivan Dokmanic. "A Graph Dynamics Prior for Relational Inference." Proceedings of the AAAI Conference on Artificial Intelligence. Vol. 38. No. 13. 2024.
>
> **W3. [Technical Contribution]** we consider ORI made several novel contributions for evolving multi-agent systems. First, we introduced a new research direction on “online relational inference for evolving multi-agent systems”. As Reviewer fSnF and Reviewer mc8m mentioned in Strengths, the proposed method is the first attempt to tackle online relational inference with a model-agnostic framework, unexplored despite wide practical applications.
>
> In addition, ORI incorporates novel adaptive learning rate technique, AdaRelation, and simple yet effective data augmentation technique, Trajectory Mirror. These two novel techniques are strategically combined with online backpropagation in the adjacency matrix-based encoder-free design, significantly enhancing the accuracy in various evolving multi-agent systems. Moreover, ORI integrates seamlessly with other NRI networks, boosting flexibility.
> We believe our technical contributions will introduce an impactful research direction, providing useful information to the related community for later research, as stated by Reviewer fSnF (Strength 3).
>
> **W4. [Limitations]** While ORI is the first attempt to learn the relational inference in online learning scenario, there have been many different problem setups in offline learning scenario. For example, more challenging environments will incorporate directed interaction graphs or/and variable number of nodes. Our current experiments do not evaluate ORI in such non-ideal environments, limiting the potential of ORI in relatively ideal environments. We will add discussion and details on the limitation of ORI in the final paper.

---

> ### Author Response · Authors · 2024-08-13
> **Follow-up from Authors**
>
> Dear Reviewer arYB,
>
> Following our previous response, we have carefully addressed all the concerns raised during the review process. We trust that our answers have adequately resolved the issues highlighted. If our answers meet your expectations, we kindly ask for your consideration in updating the final scores accordingly. Please don't hesitate to reach out if any further clarification or discussion is needed.
>
> Thank you for your time and consideration.
>
> Best regards,
>
> Authors

---

> ### Comment · Reviewer_arYB · 2024-08-13
>
> To be honest, I'm not familiar with the topic. I remain neutral with the paper. Thank you. However, I do expect more baselines. There are only four baselines now. (3 in submission and 1 in rebuttal).

---

> > ### Author Response · Authors · 2024-08-14
> > **Response to Reviewer arYB**
> >
> > We understand that the reviewer may not be familiar with this field. The authors still sincerely appreciate your efforts to understand our paper and provide insightful comments!
> >
> > We will include these discussions in the revised paper so that readers who are not familiar with the topic can easily follow the contributions of our paper. Also, we will prepare more stronger and recent baselines. Thank you once again for your thorough review.
> >
> > Sincerely,
> >
> > Authors

---

### Official Review · Reviewer_mc8m · 2024-07-04

**Soundness:** 2
**Presentation:** 2
**Contribution:** 2
**Rating:** 5
**Confidence:** 3

**Summary:**

The paper introduces a novel framework called Online Relational Inference (ORI) designed to identify hidden interaction graphs in evolving multi-agent systems using streaming data. ORI employs online backpropagation and treats the adjacency matrix as a trainable parameter, optimized through an adaptive learning rate technique called AdaRelation. Additionally, a data augmentation method named Trajectory Mirror (TM) is introduced to improve generalization by exposing the model to varied trajectory patterns. Experimental results were carried on synthetic and real-world datasets.

**Strengths:**

- The proposed method is the first addressing the dynamics in relational inference with online-learning approach.
- ORI is the first model-agnostic online relational inference framework for multi-agent systems.

**Weaknesses:**

- The reliance on an encoder-less design may not fully leverage the potential benefits of more sophisticated encoder architectures. Such as how to perform supervised learning?
- Several statements in the paper need to be explained, please refer to questions.
- The model does not explicitly address the inference of directed interactions between components.
- The evaluation metrics like AVG. Accuracy in Table 2 confused me a lot. Usually we report the final accuracy instead of the AVG. Accuracy over the entire training period.

**Questions:**

Here are my questions:

1. In line 30 "They generally perform training offline assuming the well-aligned distribution in training and testing data." Actually, I don't think this really matters, as the previous methods are unsupervised on the relations.

2. Line 40 "... primarily due to the slow optimization of the encoder." I could not reach this statement from the results in table 2. BTW, table 2 is too far from this sentence.

3. As stated: the method "employs the historical adjacency matrix to indirectly estimate the decoder’s sensitivity over the adjacency matrix and determine whether the learning needs to be accelerated." Will the errors be accumulated as well?

4. Line 145 "This means the learning is performed from the loss generated by the decoder... " Please check the ELBO loss in NRI. It contains a part called KL-term, which works on the output of encoder.

5. It would be great if the authors can elaborate more about the deviation in Eq. 1. What might be the intuition behind the deviation? How did this value change during training? Or possibly, it would be better if the equation comes with a proof on it actually works.

6. As the $I(t)$ is initially filled with 0.5. Then how to get the adjacency matrix from it to feed to the GNN?

7. What is the meaning behind the norm of gradient $\|\|\frac{dL_{mse}}{dI(t)}\|\|_1$?

8. Can you explain more about the biased data mentioned in line 200? How are these kinds of data characterized?

9. Honestly speaking, I do not get the point of Trajectory Mirror. The operations of Trajectory Mirror blurred the dynamics of the system.

10. In the experiments, how do the interaction graphs evolve? Does the performance of the proposed method correlated with the similarity between the graphs before and after change?

11. Does the method scale well to larger dynamical systems?

**Limitations:**

Although the authors state the limitation in just one sentence in the conclusion, I would like to encourage the authors discuss the limitations with more details. Such as the scalability, the adaption to directed graphs, and so on.

---

> ### Author Rebuttal · Authors · 2024-08-07
>
> **W1. [Encoder-less Design for Supervised Learning?]** This paper focuses on a fully unsupervised setup for learning the relation graph. By ‘supervised learning’ in this comment, we assume the reviewer is referring to a scenario where the true relation graph is available for the training set. We agree that having a sophisticated encoder would be beneficial in such cases, but it is not the focus of this paper.
>
> **W3. [Directed Interaction]** While the experiments in the paper do not explicitly include directed interactions, ORI does not assume any symmetry (undirected graph) in the adjacency matrix. In other words, there is no technical limitation to apply ORI in directed interaction.
>
> **W4. [Why avg Accuracy]** The interaction graph changes over time. Simply reporting the final accuracy only represents how well the model adapts to the last graph. Accordingly, we report the average accuracy over entire iterations to understand the model’s accuracy on the multiple graphs and how fast it adapts to the change in the graphs.
>
> **Q1. [Alignment in Train and Test Data]** As mentioned in line 31, the distribution in training and testing data here means not only the relations but also the governing dynamics. Temporal evolution in the dynamics, for example, if the training data is a spring system and testing data is a charged system, can significantly degrade the performance on interaction learning as studied in [1].
>
> [1] "Care: Modeling interacting dynamics under temporal environmental variation." Neurips, 2024.
>
> **Q2. [Poor Performance on Existing NRI models]** We understand that reaching this statement solely from Table 2 is not obvious. We will re-write this statement to “... significantly degrades the accuracy on relational inference since their decoder quickly learns the trajectory prediction even before the encoder generates reasonable interaction graphs.
>
> **Q3. [Error Accumulation in AdaRelation]** No. The statement refers the equation (2) (i.e., the deviation D_I(t)) which measures the norm of the difference between the current adjacency matrix and the past one. The deviation is newly measured each time step, and hence the error is not accumulated.
>
> **Q4. [ELBO Loss in NRI]** We agree that NRI has a KL-divergence loss to optimize its VAE-based model. “The learning” in line 145 we meant was supervised learning from external information, such as trajectories. We will change the statement to “The only supervision is defined by the predicted trajectories from the decoder...”.
>
> **Q6. [Initial Adjacency Matrix filled with 0.5]** Our adjacency involves continuous values as mentioned in line 166 (I_i,j(t)∈[0,1]). Hence, initial adjacency matrix filled with 0.5 is directly provided to the GNN without binarization.
>
> **Q7. [Intuition behind the Norm of Gradient |dL/dI|]** The norm of gradient indicates that how the trajectory error (∆L) changes by the adjacency matrix (∆I). Ideally, we expect this norm being high enough so that the model learns the strong correlation between the trajectory of agents and their relation. In other words, the low norm of gradient means that, the model returns the similar trajectory regardless of the relation (i.e., adjacency matrix), which is undesirable.
>
> **Q5. [Intuition behind the Deviation D_I(t)]** Our intuition in the deviation was to compare the current adjacency matrix (i.e., I(t)) with the past one (i.e., I(t-w)) to estimate whether the learning is too fast or too slow. For example, if the difference between them is large, we decrease the learning rate to stabilize the learning. Conversely, it the difference is small, we increase the learning rate to speed up the learning. We briefly provide mathematical sketch as follows: 1) expand equation (2) into ||I(t) - I(t-1) + I(t-1) - ... - I(t-w)||, 2) apply triangular inequality to get ||I(t) - I(t-1)|| + ... + ||I(t-w+1)-I(t-w)||, 3) use equation (1). This creates the inequality between the deviation and the norm of gradient, connecting the intuition behind the deviation and the norm of gradient. Also, please see Figure 3 in the attached pdf.
>
> **Q8. [Biased Learning without Trajectory Mirror]** Our data is the trajectory consists of agent's position and velocity. Let's consider a simple scenario where their position and velocity remain positive values for a while. Then, the decoder will be overfitted to positive-valued trajectories, challenging to predict negative-valued trajectory. Trajectory Mirror exposes the model to the various rotation of systems to enhance the generalizability and adaptation speed in streaming systems (e.g., 36.8% acc. increase; Table 7 in the supplementary).
>
> **Q9. [Details in Trajectory Mirror]**  Trajectory Mirror is basically a Euclidean transformation. Multi-agent systems with pairwise interactions remain invariant under Euclidean transformations when the total interaction on each agent is a sum of quasi-linear functions (e.g., agent's relative positions). This implies that Euclidean transformations, such as Trajectory Mirror, do not alter the interaction dynamics of the system, which hold true for a wide range of natural and engineered multi-agent systems [2].
>
> [1]. "Translational and rotational invariance in networked dynamical systems." IEEE Transactions on Control of Network Systems, 2017.
>
> [2]. "Consensus and cooperation in networked multi-agent systems,” Proceedings of the IEEE, 2007.
>
>
> **Q10. [How Target Relation Evolves?]** As mentioned in line 221, the target interaction graphs randomly evolve. Each simulation follows a Bernoulli process with p=0.5 for each edge in the graphs. Please see Figure 2 in the attached pdf to check the correlation between accuracy and similarity between the graphs.
>
> **Q11. [Scalability of ORI]** Please see Table 1 in the attached pdf. ORI models still outperform the existing methods in spring systems with 15 and 20 agents.
>
> **Limitations** We will elaborate the limitation of ORI as the reviewer suggested.

---

> > ### Comment · Reviewer_mc8m · 2024-08-09
> >
> > Many thanks for the rebuttal and additional experimental results. Based on the rebuttal, I have the following questions that need to be clarified with more details.
> >
> > **W1. [Encoder-less Design for Supervised Learning?] ** In my initial review, I intended to say that the previous works, like NRI, ACD and so on, consist of a clear design of encoder-decoder setup. So they can be splited into two. For example, we can perform supervised learning by just using the encoder, and we can perform simulations with just the trained decoder. So in this work, it would be hard to use the encoder separately for supervised learning.
> >
> > **W4. [Why avg Accuracy]** Could you please elaborate more on the "Iteration" mentioned in the answer? Does it refer to all iterations during training? Or just the last iteration? Some models do not learn that fast, and therefore using the Avg. Accuracy over all iterations as metric may not be fair.
> >
> > **Q1. [Alignment in Train and Test Data]** I do agree with the answer on the part of pervious works. Most of the relational inference methods do not generalize well to unseen distributions on which they are not trained. But it does not state clear the contribution of this work. In this work, the method is trained with a mixed dynamics directly, so the method does not like what is stated in this answer: trained on one distribution and then test on the other. Such as trained on springs but then tested on charged particles.
> >
> > **Q3. [Error Accumulation in AdaRelation]** Is the past adjacency matrix also learned? If it is, there would be error accumulation.
> >
> > **Q6. [Initial Adjacency Matrix filled with 0.5]** Please elaborate more on the answer. Sorry I did not get the point of how to have the input adjacency matrix as a collection of 0.5 instead of binary values to GNN? As there is no implementation or code attached, I am kind of frustrated on it.
> >
> > **Q5. [Intuition behind the Deviation D_I(t)]** What if the actual adjacency matrix changes significantly? Will it also cause large values in this term even if the learned adjacent matrix matches the actual one?
> >
> > **Q8. [Biased Learning without Trajectory Mirror]** It would be better to include how the coordinates are set in each dataset.

---

> ### Author Response · Authors · 2024-08-11
> **Response to Reviewer mc8m (1)**
>
> We also sincerely appreciate the reviewer’s time and efforts to improve the quality of our paper! The reviewer’s feedback is indeed very helpful and insightful to us. We hope our answers address your concerns during the discussion period.
>
> **W1. [Encoder-less Design for Supervised Learning?]** We agree that a clear design of encoder-decoder setup allows the separate supervised learning on the encoder. Also, we are aware of that NRI provides such codebase to supervise the encoder using cross-entropy loss. However, as we mentioned in the initial answer, our primary focus is on a “fully unsupervised setup” where the true interaction graph is *never* available to supervise the encoder. To further clarify on our problem setup, we would like to provide two primary reasons in our motivation.
>
> First, true relation graphs are generally unavailable for real-world data, where the relationship dynamics are complex and not explicitly labeled, which limits the feasibility of using supervised approaches. Second, as we focus on an online setup where the graph evolves over time, the graph should be generated in real-time as well, which is very challenging. For these reasons, the focus on our problem setup, assuming no true graph structure, is essentially different with the existing works where graph structure is assumed to be available to supervise the encoder, which we do not necessarily consider as a weakness of ORI.
>
> **W4. [Why avg Accuracy]** We would like to first clarify that, *in online learning, there is no separate training and testing stage*. The “entire iterations” in the initial answer represents all iterations during an experiment which includes 30k timesteps of observation. We expect the model to quickly identify the interaction graph at every iteration. Hence, it is undesirable if the models learn slowly, even if they reach 100% accuracy at the final iteration. Average accuracy can capture this information (i.e., how fast the model adapts).
>
> However, we agree that reporting only average accuracy may hinder the detailed understanding on the models. As the reviewer said “usually we report the final accuracy” in the initial review, we would like to provide the final accuracy on each interval (i.e., every 3k iterations) where the interaction graph remains same.
>
> *Table. Accuracy on the final iteration of each interval in springs system (10 agents) with evolving interactions*
> Method | 3k| 6k | 9k | 12k | 15k | 18k | 21k | 24k | 27k | 30k |
> ---|---|---|---|---|---|---|---|---|---|---|
> dNRI | 50.6% | 53.6% | 54.4% | 48.1% | 39.2% | 52.7% | 54.2% | 48.1% | 48.3% | 53.9% |
> MPM | 53.5% | 53.3% | 60.0% | 51.4% | 56.1% | 62.8% | 73.3% | 68.9% | 74.7% | 71.7% |
> NRI | 54.4% | 56.9% | 69.2% | 63.6% | 48.3% | 54.2% | 59.2% | 55.0% | 64.2% | 68.6% |
> ORI+NRIr | 86.7% | 96.7% | 100.0% | 94.4% | 100.0% | 100.0% | 100.0% | 100.0% | 100.0% | 100.0% |
> ORI+MPMr | 98.9% | 100.0% | 100.0% | 100.0% | 100.0% | 100.0% | 100.0% | 100.0% | 100.0% | 100.0% |
>
> The result describes that, ORI shows not only faster learning but also higher final accuracy. We appreciate your comment and will include these results in the main paper.

---

> ### Author Response · Authors · 2024-08-11
> **Response to Reviewer mc8m (2)**
>
> **Q1. [Alignment in Train and Test Data]** As stated in the above answer, in online learning, the training and testing steps are **NOT** separated. Therefore, we do not explicitly train the model on springs systems, stop training, and then test the model on charged systems.
>
> Rather, given a series of streaming observations, the model repeatedly performs online learning (no separate supervised training). Our goal is to study, whether a randomly initialized model, successfully learn a mixed dynamical system where the interaction dynamics changes with time. In our setup, the dynamics continuously changes from spring system to charge particles (and vice-versa) every 3-k iteration, as demonstrated in Figure 3(b) (main paper).
>
> The first such period of 6-k iteration, the ORI does exactly what the reviewer commented (“Such as trained on springs but then tested on charged particles”). In this period, an un-trained (randomly initialized model), only observes and learns from spring dynamics data for the first 3k iteration. At the end of 3k iterations, the (unknown) input dynamics for generating observations (data) suddenly switches to charge system. In other words, during the duration of 3k to 6k iterations, an ORI model ONLY learned spring system is being adapted on *unseen* dynamics of charge particles. As ORI continuously learns this new dynamics, it can still achieve 96.7% accuracy at the end of 6k iteration. Therefore, the first 6k iteration in Figure 3(b) exactly shows the case where the model is “trained on springs but then tested on charged particles” and performs very well.
>
> As the model continues to observe more periods of such 6k-iterations (mixed dynamics), it learns to adapt even faster to the dynamics shift. This is evident from the result in the second period of mixed-dynamics (6k-12k iterations) the model reaches 97% accuracy after 3.7k-iteration, instead of 6k-iteration in the first period.
>
> In summary, ORI can successfully adapt to the “unseen” dynamics (“trained on springs but then tested on charged particles”) without learning from mixed-dynamics data. We also see that once ORI learns from sequences of mixed dynamics (springs->charged, charged->springs), it adapts to the dynamics change faster.
>
> We will add this discussion in the main paper to clarify our contributions.
>
> **Q3. [Error Accumulation in AdaRelation]** Yes, the past adjacency matrix is also learned. We assume that, the reviewer is referring to the error in the relational accuracy. However, as we observed from experimental results in Figure 3, ORI with AdaRelation indeed showcases higher accuracy than ORI without AdaRelation (i.e., constant learning rates), which does not employ the past “learned” adjacency matrix. For example, in Figure 3(a), ORI with AdaRelation shows average accuracy of 84.8% while ORI with constant learning rates, such as relation lr 100 and relation lr 200, show 83.4% and 73.8%, respectively. Also, in Figure 3(b), the average accuracy of AdaRelation, relation lr 20, and relation lr 50 are 91.3%, 86.5%, and 90.7% respectively.
>
> We would like to clarify in AdaRelation (equation (2) & (3)), that the deviation between the current “learned” adjacency matrix and the past “learned” one is employed only for “adjusting the learning rate”, not the gradient (i.e., dL_mse/dI). We agree that unstable learning rates may influence the gradient as well in the future time steps, as in any gradient-based optimization methods, but as our adaptive learning rate in AdaRelation is bounded by equation (4), the errors due to adaptation of the learning rate are not infinitely accumulated.

---

> ### Author Response · Authors · 2024-08-11
> **Response to Reviewer mc8m (3)**
>
> **Q6. [Initial Adjacency Matrix filled with 0.5]** It’s unfortunate that the reviewer did not find our code. We have already provided the link to our codebase in the abstract of the initial manuscript. Please check the attached manuscript.
>
> The role of the adjacency matrix is to provide the graph’s edge structure to message passing layers in GNN. Instead of “vanilla message passing” where the associated edges are simply represented by binary, we consider “weighted message passing”, with the edges having continuous values. ORI performs weighted message passing from all nodes with continuous weights. Our “initial” adjacency matrix is filled with 0.5 to start with a scenario where ORI performs equally weighted message passing. As ORI performs online learning from the observations, the weights of adjacency matrix evolve according to the underlying (unknown) graph structure and dynamics.
>
> Please note, the “actual” adjacency matrices used to generate an experimental sequence is a binary one. Hence, to compute the relation accuracy, i.e., the similarity between “learned” and “actual” adjacency matrix, we threshold the “learned” matrix by 0.5 to create a binary matrix before comparing with the “actual” matrix.
>
> We will clarify ORI does weighted message passing in the main paper. Note, we do not claim “weighted message passing” as a new contribution of this paper. NRI also provides such functionality (in line 203 in train.py of NRI codebase, their edges can be either binary or continuous by a function argument “hard”).
>
> **Q5. [Intuition behind the Deviation D_I(t)]** The deviation is a function of |I(t)-I(t-w)|, where both I(t) and I(t-w) are the "learned" adjacency matrix, not the actual one. Hence, the significant change in the “actual” adjacency matrix used for generating the observed time series, may not necessarily lead to large values of the deviation. This depends on the how quickly ORI learns the “new” adjacency matrix as discussed below. We would like to remind that actual adjacency matrix is *not* available to the model for supervision.
>
> Consider a scenario when the actual adjacency matrix significantly changes between the time step ‘t-w’ and ‘t’. Assume ORI has “learned” the “actual” adjacency matrix at time step “t-w”. Now, at time step “t”, ORI can respond in two possible ways.
>
> First, ORI may quickly identify the new adjacency matrix at the time step “t”. In this case, I(t) and I(t-w) will be related to the new and previous “actual” adjacency matrices, respectively. Assuming the two “actual” adjacency matrices are significantly different, the deviation between two “learned” adjacency matrices will also be large. Hence, based on equation (3), the learning rate will decrease. This will make the learned adjacency matrix stable, thereby helping ORI to stay at the new “learned” adjacency matrix at time “t”, which is desirable as that is also the actual adjacency matrix.
>
> Let us now consider the second case where ORI does not quickly learn the new “actual” adjacency matrix and hence, the “learned” adjacency matrix at time “t” stays close to the one “learned” at time step “t-w”. In other words |I(t) - I(t-w)| remains low even if the “actual” adjacency matrices have changes. In this case, following equation (2), the learning rate increases to rapidly update the “learned” adjacency matrix, which is desirable to quickly move the “learned” matrix to the “actual” one.
>
> In summary, the equation (2) and (3) appropriately updates the learning rate when “actual” adjacency matrix changes, even without any knowledge/supervision of that “actual” matrices.
>
>
> **Q8. [Biased Learning without Trajectory Mirror]** Thank you for the suggestion. We will include it in the main paper. The coordinates of springs and charged systems are -5≤x≤5 and -5≤y≤5. The coordinates of CMU MoCap is -5.1≤x≤11.6, 0.0≤y≤29.5, and -35.2≤z≤58.2. These coordinates are same as the setup in NRI and dNRI, as we mentioned in line 220 and 227: “The synthetic datasets were generated using the open-source code from NRI” and “processed using dNRI’s open-source code”.

---

> ### Author Response · Authors · 2024-08-13
> **Follow-up from Authors**
>
> Dear Reviewer mc8m,
>
> Following our previous response, we have carefully addressed all the concerns raised during the review process. We trust that our answers have adequately resolved the issues highlighted. If our answers meet your expectations, we kindly ask for your consideration in updating the final scores accordingly. Please don't hesitate to reach out if any further clarification or discussion is needed.
>
> Thank you for your time and consideration.
>
> Best regards,
>
> Authors

---

> > ### Comment · Reviewer_mc8m · 2024-08-13
> >
> > Dear Authors,
> >
> > I apologize for my late reply, as it really took much time to get through the implementation line-by-line. I would also like to apologize for my mistake regarding the link in the paper. Many thanks for the new results and clarification of my questions.
> >
> > Yet I have another follow-up question: where does the 0.5 appear in the code? I checked both NRI and MPM in the anonymous github repo, and seems to be that neither of them explicitly shows the initial values of 0.5. By the way, what is the 'es' in the online_load_nri() of MPM? Does it contain groundtruth adjacency matrix?
> >
> > Best regards,
> >
> > Reviewer mc8m

---

> ### Author Response · Authors · 2024-08-13
> **Response to Reviewer mc8m**
>
> Dear Reviewer mc8m,
>
> We sincerely appreciate the reviewer's reply and additional questions! We are glad to know that the reviewer could find our code and provide follow-up questions.
>
> **[Where does the 0.5 appear in the code?]** initial edges are defined by “torch.ones / 2” (i.e., 0.5). For NRI, please check the line 334 in NRI/train.py, “edges = nn.Parameter( torch.ones((1, args.num_atoms*(args.num_atoms-1), 2), requires_grad=True) / 2 )”. For NRI-MPM, please check the line 385 in NRI-MPM/instructors/XNRI.py, “self.edges = nn.Parameter( torch.ones((n_atoms*(n_atoms-1),1,2), requires_grad=True) / 2 )”. We will leave the comments in the code later.
>
> **[what is the 'es' in the online_load_nri() of MPM]** We believe the reviewer is referring to ‘es’ in the line 148 in NRI-MPM/run.py. ‘es’ is to provide the node index of senders and receivers for all "possible" edges during message passing, hence *not related to the ground-truth adjacency matrix*. Please note that, we **never** use the ground-truth adjacency matrix to train the model.
>
> For example, for systems with 10 agents, ‘es’ is (2,90) matrix, where 90 represents the number of all "possible" edges excluding self-interaction (i.e., number of agents * (number of agents - 1)), and 2 represents sender and receiver node. This contains the same elements regardless of datasets:
>
>
>     [[0, 0, 0, 0, 0, 0, 0, 0, 0, - e.g., node that can send message (0-th node)
>
>     1, 1, 1, 1, 1, 1, 1, 1, 1,
>
>     2, 2, 2, 2, 2, 2, 2, 2, 2,
>
>     3, 3, 3, 3, 3, 3, 3, 3, 3,
>
>     4, 4, 4, 4, 4, 4, 4, 4, 4,
>
>     5, 5, 5, 5, 5, 5, 5, 5, 5,
>
>     6, 6, 6, 6, 6, 6, 6, 6, 6,
>
>     7, 7, 7, 7, 7, 7, 7, 7, 7,
>
>     8, 8, 8, 8, 8, 8, 8, 8, 8,
>
>     9, 9, 9, 9, 9, 9, 9, 9, 9],
>
>
>     [1, 2, 3, 4, 5, 6, 7, 8, 9,  - e.g., nodes that can receive message from 0-th node
>
>     0, 2, 3, 4, 5, 6, 7, 8, 9,
>
>     0, 1, 3, 4, 5, 6, 7, 8, 9,
>
>     0, 1, 2, 4, 5, 6, 7, 8, 9,
>
>     0, 1, 2, 3, 5, 6, 7, 8, 9,
>
>     0, 1, 2, 3, 4, 6, 7, 8, 9,
>
>     0, 1, 2, 3, 4, 5, 7, 8, 9,
>
>     0, 1, 2, 3, 4, 5, 6, 8, 9,
>
>     0, 1, 2, 3, 4, 5, 6, 7, 9,
>
>     0, 1, 2, 3, 4, 5, 6, 7, 8]]
>
> The usage of this matrix is to know the node indices of "possible" sender and receiver (e.g., for k-th edge, es[0][k] is sender and es[1][k] is receiver). We will also clarify this in the code later.
>
> As the discussion period ends soon, again, please don't hesitate to reach out if any further clarification or discussion is needed. We are always glad to answer the concerns of the reviewer.
>
> Best regards,
>
> Authors

---

> > ### Comment · Reviewer_mc8m · 2024-08-13
> >
> > Dear Authors,
> >
> > Many thanks for the response. I do not have any concerns regarding the implementation. I will raise the score to 5.
> >
> > Cheers,
> >
> > Reviewer mc8m

---

> > > ### Author Response · Authors · 2024-08-14
> > > **Response to Reviewer mc8m**
> > >
> > > Thank you for your insightful and detailed feedback throughout the rebuttal and discussion period! We are grateful for your decision to raise your score to 5. We appreciate your suggestion and will include these discussions in the revised paper.
> > >
> > > Sincerely,
> > >
> > > Authors

---

### Official Review · Reviewer_fSnF · 2024-07-05

**Soundness:** 3
**Presentation:** 3
**Contribution:** 3
**Rating:** 7
**Confidence:** 4

**Summary:**

This paper focuses on online relational inference (ORI) for dynamical systems. It points out from the optimization perspective that in the existing encoder-decoder framework, the encoder responds slowly to streaming data when inferring the evolving interaction graphs. It proposes to learn the adjacency matrix directly via a model-agnostic online learning scheme. The key design is an adaptive learning rate scheduling strategy based on the important observation that the predicted trajectory is sensitive to the change of the adjacency matrix. Delicately designed experiments show how ORI responds timely to the change of interaction graphs and dynamics, and reveal the limitations of existing methods.

**Strengths:**

S1. This paper makes the first attempt to tackle online relational inference under the streaming data setting.

S2. The observation that the predicted trajectory is sensitive to the change of the adjacency matrix is important for relational inference. It directly inspires the simple yet effective design of the online learning strategy.

S3. The carefully designed experiments help the readers understand from a mechanistic perspective how some key factors limit the inferring accuracy of existing methods, and how ORI properly overcomes these issues by its delicate design. Both the results and the analyzing procedure will be beneficial for later research.

**Weaknesses:**

W1. This paper considers the adjacency matrix as a trainable parameter. This is reasonable when the set of nodes does not change. However, with node addition and deletion, the meaning of each element in the adjacency matrix becomes inconsistent. It is unclear if ORI can handle this scenario.

W2. The trajectory mirror is indeed a data augmentation trick, although it may not be widely adopted in relational inference. I am not sure if this can be highlighted as a contribution.

**Questions:**

Q1. The setting for graph evolution can be too idealistic. The graph regularly evolves every 3k iterations, while the duration can be irregular in real-world scenarios. Will that affect the performance of the proposed method?

Q2. Do you instantiate a trainable adjacency matrix for each sample/simulation? Can ORI achieve inductive learning, i.e., generalizing to unseen samples?

Q3. How can you extend ORI to handle multiple types of interaction relations?

Q4. In the testing stage, how many iterations are required for updating the adjacency matrix at each time step? Although ORI is encoder-free, it requires at least one extra backward pass of the decoder to update the adjacency matrix. Can compare the running time of ORI with the encoder-based methods, e.g., dNRI, NRI, MPM? I believe responding in real-time is a desirable property for ORI.

---

> ### Author Rebuttal · Authors · 2024-08-07
>
> **W1. [Addition and Deletion of Node]** Thank you for pointing it out. We agree that the current ORI is studied when the number of nodes is constant. We will clarify this assumption in the revised draft, and add this as a limitation of the current study.
> We expect that ORI can be extended to scenarios when nodes are added or deleted, provided we are aware of which node is added and deleted by adding and deleting corresponding row and column in the adjacency matrix. We will update the paper to include this as a future work.
>
> **W2. [Trajectory Mirror as Contribution]** We believe that the trajectory mirror is as an effective, useful, and most importantly, simple approach to improve the accuracy. For example, Table 7 in the supplementary material, ORI with/without the trajectory mirror shows 91.3%/54.5%. Also, as the reviewer mentioned, it has not been explored before in the problem of relational inference.
>
> **Q1. [Irregular Evolution in Relations]** We consider five different cases with irregular evolutions in interaction. The system is based on springs system with 10 agents and consists of three 1k iterations, four 2k iterations, and three 3k iterations, instead of ten 3k iterations. Please see Figure 1 in the attached pdf.
>
> Case1 (Accuracy: 93.9%): interaction graph changes after 1k, 1k, 1k, 2k, 2k, 2k, 2k, 3k, 3k, 3k iterations.
>
> Case2 (Accuracy: 92.2%): interaction graph changes after 3k, 3k, 3k, 2k, 2k, 2k, 2k, 1k, 1k, 1k iterations.
>
> Case3 (Accuracy: 93.8%): interaction graph changes after 1k, 3k, 2k, 2k, 1k, 3k, 2k, 3k, 1k, 2k iterations.
>
> Case4 (Accuracy: 93.1%): interaction graph changes after 2k, 3k, 1k, 3k, 2k, 2k, 1k, 2k, 3k, 1k iterations.
>
> Case5 (Accuracy: 92.5%): interaction graph changes after 3k, 1k, 1k, 2k, 3k, 2k, 2k, 1k, 3k, 2k iterations.
>
>
> Overall, the variation in the accuracy is marginal (92.2%~93.8%), and hence, the performance of ORI is not significantly influenced by the irregular evolutions. However, since we only considered 1k, 2k, and 3k iterations, more extreme scenarios, such as few ~ thousands iterations would be an interesting future study.
>
>
> **Q2. [Instantiation of Adjacency Matrix]** We instantiate our adjacency matrix only at the initialization stage. The same adjacency matrix is used throughout the entire samples and simulations without assuming that we know when the interaction evolves. Since our online setup incorporates the streaming of unseen trajectories driven by unseen interaction graphs, system parameters, and even unseen dynamics, ORI is generalizable to unseen samples. We will clarify this in the experiment section.
>
> **Q3. [Multiple Types of Relations]** ORI is already capable of handling multiple types of interactions through a multi-channel adjacency matrix (as we mentioned in line 166, I(t) \in R^N x N x m; m is the channel dimension), where each channel represents a specific type of interaction. For instance, in the charged system, the two channels of the adjacency matrix represent attraction and repulsion, respectively. ORI determines whether there is an attraction or repulsion on an edge. Although these examples involve a small number of relation types, there is no technical limitation preventing ORI from managing scenarios where edges represent a weighted combination of multiple forces or interactions, with the possibility of zero weights for any force on a specific edge.
>
>
> **Q4. [Iterations Required to Update Adjacency Matrix]** First of all, we assume an online setup where there does not exist separate training and testing stages. Regardless of training and testing, we update the adjacency matrix only once at each time step (i.e., ORI does not perform iterative optimization).
>
> **[Computational Complexity]** We compare the overall computational complexity, including running time, in ORI with NRIr decoder and ORI with MPMr decoder with the encoder-based methods (e.g., NRI, MPM, and dNRI). In terms of trainable parameters, NRI has 721.4k for encoder and 727.3k for decoder; MPM - 1724.9k for encoder and 1071.7k for decoder; dNRI - 883.7k for encoder and 269.8k for decoder. In terms of FLOPs per iteration, NRI shows 177.8MFLOPs for encoder and 5040.5MFLOPs for decoder; MPM – 3.9GFLOPs for encoder and 10.9GFLOPs for decoder. It indicates that the decoder consumes more computation than the encoder even though they are with the similar level of trainable parameters. For the running time in a single 2080Ti GTX GPU, NRI shows 0.194sec/iter; MPM – 0.183sec/iter; dNRI – 0.096sec/iter. ORI with NRIr decoder shows 0.235sec/iter; ORI with MPMr decoder – 0.192sec/iter. While ORI shows slightly higher running time than the encoder-based models, theoretically, as ORI does not have an encoder network and apply the gradient descent to much smaller adjacency matrix, its overall FLOPs is approximated to the only decoder’s FLOPs. Also, since the FLOPs are mostly allocated at the decoder, their running time will be largely depending on the decoder design. We observe that the latency in ORI occurs at the unoptimized code for two separate gradient descent (one for the decoder and another one for the adjacency matrix). We are working on optimizing the run-time of that step in ORI.

---

> ### Comment · Reviewer_fSnF · 2024-08-08
>
> Thanks for your response and insightful discussion. The ability to infer irregularly evolving graphs with a single update at each step and slightly more running time than the decoding process is promising. Please also consider adding these discussions to the revised paper. I would tend to maintain my current rating.

---

> ### Author Response · Authors · 2024-08-11
> **Response to Reviewer fSnF**
>
> Thank you for your positive feedback and for recognizing the potential of our approach in inferring irregularly evolving graphs. We appreciate your suggestion and will include these discussions in the revised paper.
>
> We are grateful for your decision to maintain your current rating. Thank you once again for your thorough review.
>
>
> Best regards,
>
> Authors

---

### Author Rebuttal · Authors · 2024-08-07

We sincerely thank the reviewers fSnF, mc8m, arYB for their positive feedback: carefully designed experiments and analysis beneficial for later research (fSnF), the first model-agnostic online relational inference framework for multi-agent systems (mc8m), well-written with sufficient experimental validation on both synthetic and real-world datasets (arYB).

The attached pdf has three figures and two tables (reviewer fSnF: Figure 1, reviewer mcm8: Figure 2, Figure 3, Table 1, reviewer arYB: Table 2).

We hope we have addressed the reviewers’ concerns and questions regarding the paper and that they will reconsider their rating based on these discussion.

---

### Decision · Program_Chairs · 2024-09-25

**Decision:**

Accept (poster)

**Comment:**

The paper presents an interesting setting for dynamic neural relational inference where the updates are made in a streaming fashion. It proposes an adaptive learning rate updating rule for the trainable adjacency matrix with encouraging empirical results. Please add clarifications and motivations made during the discussion to the final version, especially with Reviewer mc8m.